# Position: Beyond Prediction: Toward Verifiable Physiological Waveform Reasoning with Foundation Models and Agentic LLMs

**Xiaoda Wang** [1 2]   **Ching Chang** [2]   **Defu Cao** [3]   **Kaiqiao Han** [2]   **Fang Sun** [2]   **Yue Huang** [4]   **Minxiao Wang** [5]
**Chang Xu** [6]   **Xiao Luo** [7]   **Runze Yan** [5]   **Xiangliang Zhang** [4]   **Xiao Hu** [5]   **Yan Liu** [3]   **Yizhou Sun** [2]   **Wei Wang** [2]
**Carl Yang** [1]

## Abstract

Physiological waveforms (e.g., ECG, PPG, EEG) encode clinically meaningful information in fine-grained morphology, precise timing, and cross-channel dynamics, yet most machine learning systems still treat them as generic time series and optimize end-to-end prediction. In this position paper, **we argue for verifiable physiological waveform reasoning: extracting localized, measurable signal evidence from raw signals, interpreting that evidence into physiological semantics, and supporting clinically grounded decisions.** Waveform reasoning is challenging due to acquisition heterogeneity, signal fidelity, complex semantics and cross-channel coupled dynamics. We analyze why existing model families remain insufficient: physiological foundation models learn strong perceptual representations but remain weak at verifiable reasoning, while LLM-based adaptations have limited waveform understanding. To bridge this gap, **we advocate verifiable, closed-loop systems that unify waveform semantics with language intelligence.** Concretely, we propose a dual-process architecture that System 1 aligns physiological waveforms with language, and System 2 provides agentic reasoning via a Plan–Act–Verify loop, together enabling verifiable physiological waveform reasoning. And we propose evaluations beyond accuracy, emphasizing traceability, replayability, counterfactual robustness, and calibrated abstention.

[1]Department of Computer Science, Emory University [2]Department of Computer Science, University of California, Los Angeles [3]Department of Computer Science, University of Southern California [4]Department of Computer Science, University of Notre Dame [5]Nell Hodgson Woodruff School of Nursing, Emory University [6]Microsoft Research [7]Department of Statistics, University of Wisconsin–Madison. Correspondence to: Xiaoda Wang <xiaoda.wang@emory.edu>.

*Proceedings of the $43^{rd}$ International Conference on Machine Learning*, Seoul, South Korea. PMLR 306, 2026. Copyright 2026 by the author(s).

## 1. Introduction

Physiological waveforms such as electrocardiograms (ECG), photoplethysmograms (PPG), and electroencephalograms (EEG) are high-fidelity, time-resolved measurements of underlying biological dynamics. Unlike many time-series forecasting benchmarks where performance can often be driven by coarse trends, seasonality, or global summary statistics (Chang et al., 2025a; Wang et al., 2026a; Ye et al., 2025; Liu et al., 2025c; Jia et al., 2024; Cao et al., 2024; Ye et al., 2026), the clinical meaning of physiological waveforms is concentrated in fine-grained morphology, precise timing, and structured dependencies across channels. Localized events (e.g., QRS onsets/offsets, dicrotic notches) support standardized measurements and guideline-driven interpretation (Clifford et al., 2012; Wang et al., 2026c; Orphanidou et al., 2014; Wang et al., 2026b; Jin et al., 2025; 2026). In practice, clinicians do not "read" a waveform by a single global score; they delineate events, measure intervals and amplitudes, assess signal quality, and reconcile inconsistencies across leads or modalities before reaching guideline-constrained decisions (Shcherbina et al., 2017; Bent et al., 2020).

Recent years have seen rapid progress in applying deep learning to waveform modeling, achieving strong performance in arrhythmia detection and broader ECG interpretation (Hannun et al., 2019; Ribeiro et al., 2020; Attia et al., 2019; Ismail Fawaz et al., 2019). More recently, physiological foundation models (PhysioFMs) promise transferable representations and scalable pretraining regimes (Wiggins & Tejani, 2022; Mehari & Strodthoff, 2022; Li et al., 2025b). In parallel, large language model (LLM)-centric adaptations have emerged that use language as an interface for clinical tasks, including explanation generation, and guideline-aware summarization. Despite these advances, much of the literature still emphasizes end-to-end prediction. This clinician workflow also highlights a failure mode that end-to-end prediction can obscure: undetected wrongness under acquisition artifacts and heterogeneity. A motion-corrupted PPG segment can mimic irregular rhythm; a baseline wander can distort ST segments; missing or swapped ECG leads can

yield plausible-looking signals that nevertheless invalidate downstream measurements. A system that outputs a prediction without exposing what evidence it relied on is difficult to audit.

**We argue that physiological signal AI should be framed as physiological waveform reasoning: extracting localized, verifiable evidence from raw signals, translating that evidence into physiological semantics, and producing clinically grounded decisions rather than only end-to-end predictions.** Here, "verifiable" is an operational requirement that reasoning be entailed by measurable evidence objects. The decision should be creditable only to the extent that it can be reconstructed from recorded evidence objects and the procedures that produced them, yielding traceability and replayability rather than post-hoc narrative.

This reframing also clarifies why current model families remain insufficient. PhysioFMs excel as perceptual backbones, but their typical outputs do not constitute evidence objects with provenance; they provide limited mechanisms for unit-consistent measurements, explicit physiological semantics, or guideline-constrained decision logic (Rudin, 2019; Li et al., 2025b). Conversely, LLM-centric adaptations can improve the linguistic form of reasoning (e.g., producing fluent explanations, summaries, or differential diagnoses) but often rely on lossy interfaces to high-frequency morphology, weak grounding to raw signals, and rationales that are not guaranteed to be logically supported by the waveform (Schick et al., 2023; Shinn et al., 2023; Cao et al., 2024; Goswami et al., 2024; Jin et al., 2024b).

**We therefore advocate verifiable, closed-loop systems that unify waveform semantics with language intelligence.** Concretely, we propose a dual-process that *System 1* aligns the model with physiological waveforms and language, enabling faithful waveform understanding, while *System 2* performs agentic reasoning via a Plan–Act–Verify loop that decomposes tasks, requests missing evidence, invokes deterministic measurement and validation tools, checks cross-view consistency, and abstains or escalates when evidence quality or constraints do not support a safe conclusion (Guo et al., 2017; Ovadia et al., 2019; Geifman & El-Yaniv, 2019). Accordingly, evaluations should move beyond accuracy to score whether outputs are traceable to localized evidence, replayable under logged procedures, robust to nuisance perturbations yet sensitive to clinically meaningful counterfactual changes, and uncertainty-aware through calibrated abstention or escalation.

**Contributions.** The main contributions are as follows: ❶ We reframe physiological signal AI as *verifiable physiological waveform reasoning*: extracting evidence from raw signals, mapping evidence to physiological semantics, and supporting clinical decisions. ❷ We analyze the key challenges and why current model families remain insufficient.

PhysioFMs rarely expose auditable verification objects, while LLM-centric adaptations often rely on lossy waveform grounding. ❸ We propose a dual-process, closed-loop blueprint in which *System 1* aligns physiological waveforms with language, while *System 2* performs agentic Plan–Act–Verify reasoning for meaningful physiological waveform reasoning. ❹ We advocate evaluations beyond end-to-end prediction, emphasizing evidence traceability, replayability, counterfactual robustness, and uncertainty-aware decisions.

## 2. Beyond Prediction: Toward Verifiable Physiological Waveform Reasoning

### 2.1. What are Physiological Waveforms

Physiological waveforms encompass the direct measurement of the body's electrical activity or hemodynamic responses. We define physiological waveforms as high-fidelity temporal projections of continuous biological processes. Unlike generic time-series data (e.g., financial stocks or weather metrics) where analysis often centers on global trends, seasonality, or statistical distribution (Chang et al., 2025a; Ye et al., 2025; Chang et al., 2025c), physiological waveforms are characterized by precise morphological semantics and mechanistic coupling. The information is dense and encoded not just in the value at time $t$, but in the morphology, phase relationships, and quasi-periodic structure. Formally, we represent a physiological waveform recording as a multivariate tensor $\mathbf{X} \in \mathbb{R}^{C \times T}$, where $T$ represents the discrete sampling of a continuous biological state and $C$ denotes the spatial or modal channels (e.g., 12 leads of ECG). We use physiological waveforms to refer to widely used sensing modalities such as ECG, PPG, EEG, EMG, and PCG. For completeness, we summarize their sensing principles and typical characteristics in Appendix A.

### 2.2. Verifiable Physiological Waveform Reasoning

We define *physiological waveform reasoning* as the capability to transform raw biosignals into actionable clinical logic through a structured inference process grounded in waveform evidence. Unlike standard end-to-end prediction, which may rely on opaque correlations (Ismail Fawaz et al., 2019; Rudin, 2019), reasoning requires explicit intermediate products that connect low-level signal dynamics to high-level physiological concepts under established clinical principles (Rajpurkar et al., 2022; Wagner & Strauss, 2013).

We treat *verifiability* as an operational contract. Specifically, a system should emit *verification objects*—signal-quality summaries, localized events with explicit time/lead indices, and unit-consistent measurements defined by reproducible windows and procedures—so that intermediate claims and final decisions are acceptable only if they can be re-derived by replayable checks. This yields an auditable interface

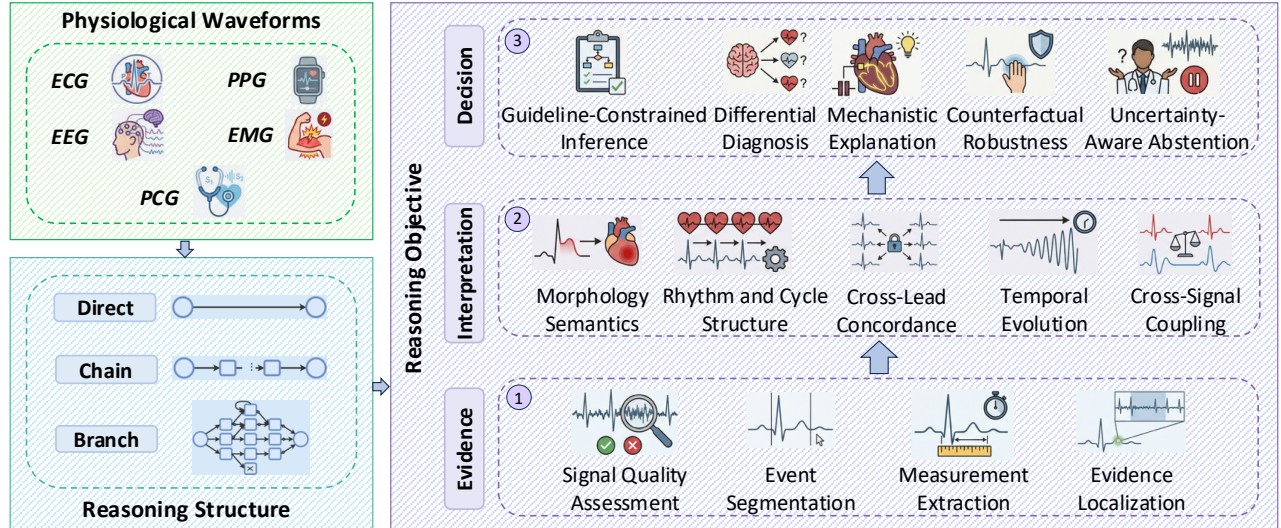

*Figure 1.* **Verifiable physiological waveform reasoning.** We organize reasoning along two axes: (1) **Left: Reasoning Structure** describes the inference topology: *Direct*, *Linear-Chain*, or *Branch-Structured*. (2) **Right: Reasoning Objective** specifies the target level: Level 1 *Evidence* (e.g., segmentation/measurement), Level 2 *Interpretation* (e.g., semantics), and Level 3 *Decision* (e.g., diagnosis).

where conclusions remain grounded in localized, measurable waveform evidence rather than post-hoc narrative. We organize physiological waveform reasoning along two axes (Fig. 1): *Reasoning Structure* and *Reasoning Objective*

**Reasoning Structure.** We define *Reasoning Structure* as the topology by which a system composes intermediate states into a conclusion (Wei et al., 2022). We summarize three topologies: *Direct* reasoning, *Linear-chain* reasoning, and *Branch-structured* reasoning. In practice, many systems either adopt a single topology or are hybrids that combine multiple topologies.

*(1) Direct Reasoning.* A single-pass mapping from raw waveforms directly to a clinical output without explicit intermediate states. This is the dominant paradigm in current physiological deep learning (e.g., end-to-end arrhythmia detection (Hannun et al., 2019; Attia et al., 2019)). While effective for pattern matching, this "black-box" approach lacks the mechanisms to isolate specific morphological evidence (e.g., P-wave absence), making it difficult to distinguish between true pathology and artifacts.

*(2) Linear-Chain Reasoning.* A sequential inference process where conclusions are built through intermediate, checkable steps (e.g., *Event Delineation → Interval Measurement → Rule Application → Diagnosis*). Adapting the principles of Chain-of-Thought prompting (Wei et al., 2022) to waveforms, this structure mimics the standard clinical workflow of first quantifying indices before interpreting them, thereby enforcing logical traceability and allowing for error localization at specific steps (Koh et al., 2020).

*(3) Branch-Structured Reasoning.* A non-linear process

that explores multiple concurrent hypotheses or verification paths before aggregating a final decision. Inspired by Tree-of-Thoughts frameworks (Yao et al., 2023), this approach is uniquely suited for complex differential diagnosis. For example, a system might spawn separate reasoning branches to evaluate competing explanations for a wide-complex tachycardia (e.g., *Branch A: VTach vs. Branch B: SVT with Aberrancy*), weighing the evidence for each hypothesis against clinical guidelines before converging on a conclusion.

**Reasoning Objective.** We define *Reasoning Objective* as the level of capability a system is expected to achieve. We propose a hierarchy of increasing complexity: *Level 1: Evidence*, *Level 2: Interpretation*, and *Level 3: Decision*.

*(1) Level 1: Evidence.* This level is to extract verifiable observations from raw waveforms. It prioritizes auditable outputs (timestamps and unit-bearing measurements) that can be independently re-computed from the signal.

❶ *Signal Quality Assessment.* This objective characterizes whether a segment/lead is physiologically trustworthy by detecting acquisition failures and artifacts and reporting explicit quality indicators (e.g., SQIs) used for weighting or exclusion (Clifford et al., 2012; Orphanidou et al., 2014).

❷ *Event Segmentation.* This objective anchors clinically defined fiducial points with precise time indices (e.g., P/QRS/T onsets/offsets for ECG) under standardized conventions, providing the coordinates required for reproducible measurement (Party et al., 1981; Chang et al., 2025b;c).

❸ *Physiological Measurement Extraction.* This objective derives clinical indices as scalar values with physical units

and well-defined windows (e.g., QT/QTc, QRS duration, HRV, PTT), ensuring results are recomputable from fiducials rather than implicit latent states (Malik, 1996; Ding & Zhang, 2019).

❹ *Evidence Localization.* This objective links every claim to the exact time ranges and channels/leads (and intermediate fiducials/measurements) that support it, enabling auditability and discouraging unsupported post-hoc narratives (Rudin, 2019).

*(2) Level 2: Interpretation.* This level synthesizes Level-1 evidence into physiological semantics, turning timestamps and measurements into clinically meaningful concepts under biological constraints.

❶ *Morphology Semantics.* This objective maps waveform geometry to physiological and pathological concepts (e.g., sawtooth atrial activity → atrial flutter; ST elevation → acute myocardial injury/ischemia). The emphasis is concept grounding in standardized ECG interpretation conventions rather than pattern matching alone (Kligfield et al., 2007).

❷ *Rhythm and Cycle Structure.* This objective infers the organizing logic of cycles from event sequences (e.g., regularity, bigeminy, compensatory pauses), using interval patterns and beat-to-beat dependencies. It is distinct from morphology because the same beat shape can appear under different rhythm regimes (Moody & Mark, 2001).

❸ *Cross-Lead Concordance.* This objective enforces spatial consistency across leads as multiple views of the same cardiac source (e.g., inferior MI patterns coherently appearing in II/III/aVF but not aVL). It also uses multi-lead redundancy to separate global physiology from lead-specific corruption (Surawicz et al., 2009).

❹ *Temporal Evolution.* This objective reasons over trajectories rather than snapshots, capturing clinically meaningful state transitions (e.g., progressive QRS widening, evolving repolarization changes). The key output is a time-ordered narrative of change supported by repeated evidence over windows (Goldberger et al., 2000).

❺ *Physiological Cross-Signal Coupling.* This objective enforces mechanistic coherence across signals (e.g., an ECG electrical event should be followed by a mechanical pulse in PPG/ABP within a plausible latency such as PTT/PAT). Violations are treated as evidence of misalignment, or sensor failure rather than physiological conclusions (Allen, 2007).

*(3) Level 3: Decision.* This level converts interpreted evidence into actionable clinical logic, including rule-based conclusions, hypothesis management, and risk-aware decision behavior.

❶ *Guideline-Constrained Inference.* This objective applies explicit clinical criteria to Level-1 measurements with trans-parent rule invocation, especially for borderline cases. Outputs should explicitly cite which guideline conditions were satisfied or violated (Surawicz et al., 2009).

❷ *Differential Diagnosis.* This objective generates and ranks competing hypotheses that can explain the same evidence (e.g., distinguishing *VT* from *SVT with aberrancy*). Ranking should reflect supporting versus contradicting evidence for each hypothesis (Reiter, 1987).

❸ *Mechanistic Explanation.* This objective provides a physiology-grounded causal rationale for the conclusion (e.g., PR prolongation indicating delayed AV nodal conduction). The goal is a plausible causal account that connects observed measurements to underlying mechanisms (Neuberg, 2003).

❹ *Counterfactual Robustness.* This objective stress-tests decision stability under realistic perturbations, identifying which evidence is necessary versus incidental. Robustness is treated as a verification step rather than a post-hoc justification (Wachter et al., 2017).

❺ *Uncertainty-Aware Abstention.* This objective calibrates confidence and supports safe deferral when evidence quality or ambiguity is high (e.g., requesting re-recording instead of forcing a label). Abstention should be principled (risk–coverage tradeoff), not ad hoc (Kompa et al., 2021).

## 3. Why Current Model Families Fall Short

### 3.1. Key Challenges

**Challenge 1: Acquisition Heterogeneity and Signal Fidelity.** Acquisition heterogeneity directly degrades signal fidelity: intermittent wear (battery/adherence gaps), motion/perfusion artifacts, and device-specific sampling/lead configurations can systematically warp waveform morphology and timing rather than producing rare "outliers" (Kligfield et al., 2007; Shcherbina et al., 2017; Bent et al., 2020; Orphanidou et al., 2014; Clifford et al., 2012; Fine et al., 2021). Reasoning models must therefore infer and condition on acquisition state (quality, continuity, configuration) as an explicit latent variable before clinical interpretation (Orphanidou et al., 2014; Clifford et al., 2012; Han et al., 2024; Wang et al., 2025c).

**Challenge 2: Complex Physiological Semantics.** Waveform "language" is encoded in localized, high-frequency morphology (e.g., J-point/ST–T shape, dicrotic notch) that must be preserved and mapped to physiological concepts (Wagner & Strauss, 2013; Kligfield et al., 2007). Bridging pattern to mechanism requires grounding measurements in guideline definitions (e.g., ST elevation and acute myocardial injury/ischemia) rather than relying on geometric similarity alone (Thygesen et al., 2018).

**Challenge 3: Multivariate and Cross-Channel Dynamics.** Physiological signals are coupled views of shared biology (e.g., ECG electrical activation preceding the hemodynamic PPG pulse), imposing tight phase/latency constraints across channels (Allen, 2007; Orphanidou et al., 2014). Effective reasoning must test cross-channel concordance to separate true pathology from single-sensor corruption under motion-related noise (Fine et al., 2021; Clifford et al., 2012).

**Challenge 4: The Data-Reasoning Mismatch.** Most public datasets provide coarse labels without intermediate evidence (e.g., measurements, rule traces), limiting supervision for verifiable inference (Wagner et al., 2020; Johnson et al., 2023; Gow et al., 2023; Oh et al., 2023). In contrast to NLP where explicit rationales (e.g., chain-of-thought) are common, waveform reasoning needs evidence-centric representations that are both inspectable and intervention-friendly (Wei et al., 2022; Koh et al., 2020; Rudin, 2019).

### 3.2. Model Families and Limitations

We categorize prior work by model family because architecture determines the evidence interface and the resulting justification, spanning physiological foundation models (Section 3.2.1) and LLM-centric pipelines (Section 3.2.2).

#### 3.2.1. PHYSIOLOGICAL FOUNDATION MODELS

PhysioFMs are shifting from task-specific supervised pipelines to reusable biosignal backbones pretrained with self-supervision (Yang et al., 2023; Jiang et al., 2025b; Kataria et al., 2025; Xu et al., 2025b). One route centers on scaling data and external validation, producing open or broadly accessible backbones and cross-domain evaluations (McKeen et al., 2025; Xu et al., 2025c; Li et al., 2025b; Abbaspourazad et al., 2023; Luo et al., 2024; Xu et al., 2025a; Saha et al., 2025; Cao et al., 2026b). Second route is to better encode physiological structure: self-supervised designs that preserve spatio-temporal dependencies and yield broad downstream utility (Coppola et al., 2024; Na et al., 2024; Wang et al., 2025b). A third route pushes representations toward richer clinical semantics and unified multi-task interfaces, sometimes via diagnosis or disease-centric objectives, and via language-aligned or multi-task waveform modeling (Tian et al., 2024; Jiang et al., 2024; 2025b; Cui et al., 2024). Finally, multimodal and time–frequency pretraining is emerging (notably in sleep/PSG), while cross-modal guidance and new sensing modalities broaden the scope of what "physiological foundation models" cover (Thapa et al., 2026; Huang et al., 2026; Kjaer et al., 2025; Pillai et al., 2025; Nie et al., 2025; Chen et al., 2025; Zhang et al., 2024; 2023).

**Limitations Analysis.** PhysioFMs primarily strengthen Level-1 (Evidence) capability by learning robust waveform representations under noise and domain shift. However, they rarely support complex physiological waveform reasoning because their dominant interface remains prediction-centric rather than evidence-centric. So they do not reliably expose localized evidence, translate morphology into intermediate interpretations, or implement explicit decision procedures with verification and uncertainty-aware abstention.

#### 3.2.2. LLM-CENTRIC ADAPTATIONS

LLM-centric methods are increasingly organized around a language-grounded interface. Rather than producing predictions alone, recent systems align waveforms with text and leverage multimodal instruction tuning to generate reports, answer questions, and support interactive interpretation (Zhao et al., 2025b; Wan et al., 2025; Yang et al., 2025; Cao et al., 2026a; Zhang et al., 2025; Yang et al., 2026). Building on this interface, a growing line of work strengthens ECG–text alignment with clinically informed supervision, improving semantic grounding and generalization (Yu et al., 2024; Liu et al., 2025a; Li et al., 2025a; Weng et al., 2026). In parallel, "ECG understanding" and "clinical reasoning" are made more measurable by reframing evaluation as clinically oriented question answering, supported by knowledge-informed multimodal QA protocols (Oh et al., 2023; Wang et al., 2025a; Xie et al., 2025b; Pham et al., 2025; Xie et al., 2024). To further improve factuality and traceability at inference time, many pipelines attach external clinical knowledge and retrieval (RAG-style) to report generation and diagnosis/QA (Tang et al., 2025; Yu et al., 2023; Cao et al., 2025). Finally, several efforts move toward more LLM-native signal interfaces by mapping waveforms into language-compatible units and bridging EEG–language for open-vocabulary decoding and assisted documentation (Jiang et al., 2025b; Chan et al., 2025; Jiang et al., 2025a).

**Limitations Analysis.** Many LLM-centric works still frame the task as question answering or answer prediction and thus rarely perform complex reasoning. Moreover, their waveform understanding is often shallow and indirect: upstream feature extraction can be lossy, the mapping from representations back to measurable evidence is frequently opaque, and generated rationales may be fluent but not entailed by the waveform under artifacts, missing channels, or distribution shift (Huang et al., 2025). Without explicit verification loops, these systems rarely guarantee guideline-consistent decisions or uncertainty-aware abstention.

## 4. Framework Design: Unifying Waveform Semantics and Language Intelligence

### 4.1. Design Goals

To bridge physiological signals and symbolic logic, we distill three design goals for waveform reasoning systems. (1) ***Joint waveform understanding and language-level rea-***

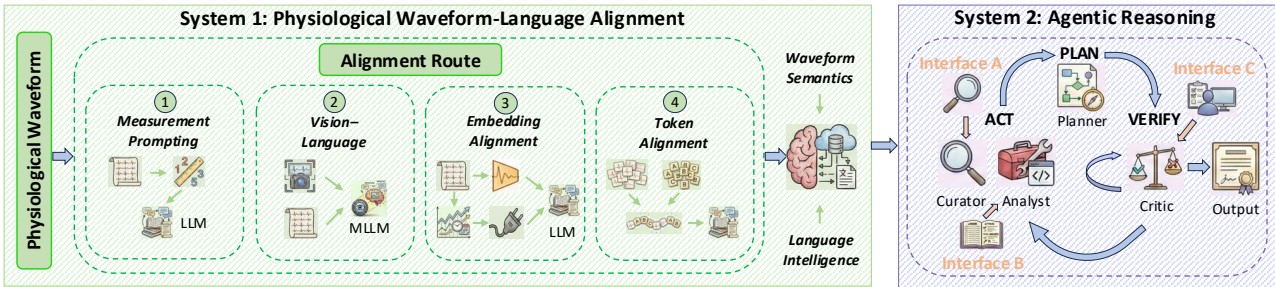

*Figure 2.* **Framework Design: Unifying Waveform Semantics and Language Intelligence.** System 1 aligns physiological waveforms and language via four alignment routes; System 2 performs plan–act–verify reasoning with tool grounding and human oversight.

*soning.* The system must align physiological waveform with language to have strong reasoning ability and waveform understanding ability (Nie et al., 2023). (2) *Agentic reasoning for complex reasoning objectives.* The system must iteratively test hypotheses via targeted measurements, cross-lead consistency checks, and tool-grounded computation (Yao et al., 2023; Shinn et al., 2023). (3) *Human-centered closed-loop evaluation and oversight.* The system must provide auditable verification objects, abstain or escalate under uncertainty, and improve via feedback-driven evaluation (Shinn et al., 2023; Yao et al., 2023).

### 4.2. Dual-Process Framework Architecture

Inspired by the dual-process theory in cognitive science (Kahneman, 2011) and its recent adaptations in machine learning (Bengio, 2017; Goyal & Bengio, 2022), we advocate a *Dual-Process Architecture* with two systems to satisfy these goals simultaneously. System 1 and System 2 are **functionally distinct but operationally coupled**: (1) *System 1 (Physiological Waveform-Language Alignment):* this system defines the alignment mechanism between physiological waveforms and the language model, so the system attains both strong reasoning ability and strong waveform understanding. (2) *System 2 (Agentic Reasoning: Plan–Act–Verify):* this system is the controller that performs agentic reasoning. It plans, acquires missing verification objects via System 1 and tools, verifies claims with deterministic measurements, checks physiological/guideline constraints, and abstains or escalates under uncertainty.

#### 4.2.1. SYSTEM 1: PHYSIOLOGICAL WAVEFORM–LANGUAGE ALIGNMENT

The central question for System 1 is: *how should waveforms be aligned with a language model so that the system achieves strong reasoning ability and waveform understanding?* We propose **four alignment routes**, distinguished by the interface granularity and the degree of coupling:

**(i) Measurement Prompting Alignment.** The most simple route is to not align raw waveforms at all, but instead

align waveform measurements into structured prompts that an LLM can reliably read. Recent works perform few-shot prompting on physiological time series by directly serializing measured sequences into the prompt (Liu et al., 2023), and propose retrieval-augmented, measurement-driven prompting for ECG diagnosis (Yu et al., 2023). In the general time-series domain, prompt-based reprogramming of LLMs (treating time series as a foreign language via prompt design) (Liu et al., 2024a; Kong et al., 2025; Chang et al., 2025d; 2024b) also fits this route when the interface is primarily textual verification objects rather than learned waveform embeddings.

**(ii) Vision–Language Alignment.** Vision–Language alignment converts waveforms into a visual surrogate and then leverages the well-developed MLLM stack for reasoning. This matches how humans consume waveforms ("look at the tracing"), enabling direct reuse of vision–language training recipes and instruction tuning. Recent work shows strong potential for ECG-image instruction tuning and benchmarking (Liu et al., 2024b), and for grounded ECG understanding by combining plots with additional modalities or verification objects (Lan et al., 2025; Seki et al., 2025).

**(iii) Embedding Alignment.** Embedding alignment feeds the LLM continuous vectors computed from the raw waveform, retaining more signal detail. There are two common sub-routes. (1) *Specialized waveform encoders*: train an encoder jointly with an LLM-facing interface under reasoning supervision (instruction tuning, or contrastive alignment to clinical text), so the representation is optimized for downstream reasoning rather than generic reconstruction (Chow et al., 2024; Jin et al., 2024a; Langer et al., 2025; Chang et al., 2024a). (2) *Adapter-based embedding alignment*: freeze a strong pretrained PhysioFM encoder and learn lightweight adapters to map its representations into the LLM's embedding space (Yu et al., 2025). The advantage is compute efficiency, as well as better retention of raw morphology than rendering.

**(iv) Discrete Token Alignment.** Token alignment makes waveform a *native language* by discretizing signal patches

into tokens that are processed by an autoregressive LLM. This enables the cleanest conceptual integration—signals and text become a single sequence model—and opens the door to true native multimodal chain-of-thought reasoning, where the model can attend directly to signal tokens. Examples include large-scale tokenization-based pretraining that frames forecasting as language modeling (Ansari et al., 2024), and wavelet-based tokenization that discretizes time-localized frequency coefficients for autoregressive forecasting (Masserano et al., 2025).

### 4.2.2. SYSTEM 2: AGENTIC REASONING: PLAN–ACT–VERIFY

System 2 treats waveform interpretation and decision-making as an iterative Plan–Act–Verify procedure rather than a single forward pass from inputs to labels. The reasoner plans, acts by requesting additional verification objects or invoking external procedures, and verifies intermediate claims before committing (Yao et al., 2023; Shinn et al., 2023). This agentic loop is especially important for complex waveform reasoning, where correctness depends on iterative evidence acquisition, cross-checking, and principled backtracking under uncertainty (Wei et al., 2026; Zhao et al., 2025a; Liu et al., 2025b; Xie et al., 2025a).

Crucially, reasoning quality is measured by the trajectory: which verification objects are requested, which tools are used, how contradictions are handled, and when the system stops. System 2 maintains an explicit working state and selects actions that reduce uncertainty with minimal cost, terminating only when the decision is supported by auditable verification objects and physiological or guideline constraints are satisfied; otherwise it triggers targeted re-measurement or evidence expansion (Shinn et al., 2023). To make this loop auditable, we propose a minimal **Role-based Agentic Architecture** with a Curator (evidence seeking), an Analyst (deterministic execution), and a Critic (verification).

**(i) The Curator: hypothesis-driven active perception (Act).** Most models passively accept a fixed input window, failing when decisive verification object lies outside the receptive field or is obscured by artifacts. The Curator addresses this by steering System 1 toward *targeted verification object acquisition*: it requests additional temporal context, alternative lead subsets, or focused re-representations conditioned on the current hypothesis (e.g., *"Retrieve the preceding 30 seconds to test sudden vs. gradual onset"*) (Zhao et al., 2025a). This active perception mirrors clinical workflows in which clinicians zoom, scroll telemetry, or examine specific leads before declaring a rhythm or morphology-based diagnosis.

**(ii) The Analyst: deterministic tool execution with provenance (Act → Verify).** Waveform reasoning often hinges on precise measurements, which LLMs are not reliable at producing directly. The Analyst therefore implements *code-as-reasoning*: instead of emitting numbers as free-form tokens, it calls deterministic tools or executable code for delineation, peak detection, interval computation, and statistical checks, ensuring each quantitative claim is backed by a reproducible execution trace (provenance) rather than a stochastic guess (Liu et al., 2025b). This also enables principled re-measurement when the Critic requests alternative windows, preprocessing assumptions, or robustness checks.

**(iii) The Critic: reflexion, constraint checking, and guideline adherence (Verify).** A major failure mode of end-to-end systems is producing fluent but physiologically impossible claims. The Critic implements a reflexion loop that stress-tests intermediate conclusions against (i) physiological constraints (e.g., rhythm regularity vs. RR variability, plausible interval ranges, cross-lead consistency) and (ii) guideline logic when applicable. When inconsistencies are detected (e.g., *"irregular rhythm"* declared but RR intervals are constant), the Critic triggers *backtracking*: it revises the plan, asks the Curator for additional verification object, and instructs the Analyst to re-measure or run alternative checks (Shinn et al., 2023; Liu et al., 2025b). Importantly, the Critic should also control *stop conditions*: if verification object quality remains low or tool outputs conflict, it should abstain or escalate rather than force a brittle decision.

**Closing the Loop: Verification, Provenance, and Oversight Interfaces.** While the Curator–Analyst–Critic decomposition specifies who acts in the Plan–Act–Verify loop, verifiable waveform reasoning requires explicit *closed-loop interfaces* that specify *what* is exchanged and recorded. **Interface A: Verification object querying from System 1.** System 2 can call System 1 to obtain alternative waveform-grounded views under uncertainty. Concretely, System 2 may request different temporal context, lead subsets, preprocessing assumptions, or specific verification objects (e.g., delineation- or measurement-oriented outputs). Importantly, this is not model updating; it is inference-time, hypothesis-driven verification object acquisition over a fixed alignment mechanism. **Interface B: Auditable ledger of verification object for tool-grounded claims.** Rather than reiterating tool use, we require a protocol-level verification object ledger: any quantitative assertion produced by System 2 can be accompanied by a replayable record of how it was obtained (tool name, parameters, software version, input segment/lead identifiers, and failure modes when applicable). This ledger makes outputs reproducible and supports debugging and downstream auditing, while leaving tool selection and execution to the Analyst behavior. **Interface C: Human oversight and feedback-to-memory.** For safety-critical deployment, the system should expose auditable artifacts so clinicians can validate what the model relied on and intervene when verification object quality is low. Crucially, human signals should be captured in structured form

and stored as workflow memory. Over time, these feedback traces can be retrieved to guide future Plan–Act–Verify trajectories or internalized through post-training optimization, enabling continual improvement beyond one-off reflexion.

# 5. Evaluation: From Prediction to Verifiable Physiological Waveform Reasoning

Physiological waveform reasoning should be evaluated as **verifiable, episode-level decision making**, not single-shot prediction. The goal is not only to be correct, but to be *checkable*: intermediate claims and final decisions are credited only if they can be re-derived from reproducible **verification objects** and replayable procedures, with uncertainty handled via abstention or escalation when support is insufficient.

## 5.1. Episode-Level Evaluation via a Verification Ledger

**Unit (*reasoning episode*).** We evaluate a *reasoning episode*: the complete trajectory from waveform input and task query to final output, including intermediate requests for additional views, verification-object extraction, measurements, tool calls, and any backtracking. The episode is the atomic unit.

**Interface (*Verification Ledger*).** To make verifiability testable, each episode must expose a *Verification Ledger*: a minimal structured record that links (i) *verification objects:* signal-quality summaries, localized events with explicit time/lead indices, and unit-consistent measurements defined by reproducible windows and procedures; (ii) *tool provenance:* tool names, parameters, software versions/hashes, input segment/lead identifiers, and failure modes; and (iii) *the final decision:* including abstention or escalation when warranted. A decision is credited only if it can be reconstructed from the ledger by replayable checks.

**Criteria and metrics.** Given the ledger interface, we score task-agnostic episode-level criteria: (i) *Traceability*: major claims and decision labels are linked to sufficient verification objects or tool outputs for audit; (ii) *Replayability*: quantitative statements are reproducible by rerunning the logged procedures under recorded parameters; (iii) *Robustness vs. sensitivity*: outputs remain stable under nuisance perturbations (noise, baseline wander, resampling, preprocessing variants, missing leads) yet respond appropriately to clinically meaningful counterfactual changes; (iv) *Uncertainty-aware safety*: the system surfaces low-quality signals, conflicts, or violated constraints and abstains/escalates when a safe conclusion is not supported. And *budgeted verifiable success* should be additionally reported: performance as a function of query/tool-call budget and audit effort, rather than a single unconstrained score.

## 5.2. Component and Closed-Loop Evaluation

**Component evaluation.** In the dual-process framework, System 1 is evaluated by *verification-object fidelity*: correctness of extracted signal-quality summaries, localized events, and unit-consistent measurements, consistency across views, and calibration of quality indicators under heterogeneity. System 2 is evaluated by *trajectory validity*: whether it requests appropriate verification objects, applies coherent checks and constraints, ensures intermediate claims and final decisions are entailed by the ledger, detects and resolves contradictions via principled backtracking, and invokes abstention/escalation when warranted.

**Closed-loop Interfaces.** System-level evaluation tests whether closed-loop integration improves verifiable outcomes under realistic constraints. **Interface A: Verification-object querying** compares single-shot episodes to budgeted multi-turn episodes where System 2 can query System 1 for alternative views, reporting improvement in budgeted verifiable success and reduction in unresolved conflicts per additional query. **Interface B: Auditable Verification Ledger** measures replay consistency, mismatch rate, and conflict handling. **Interface C: Audit and correction** quantifies audit cost and correction yield, with special focus on abstention under low-quality signals, where safe deferral is preferable to confident but uncheckable conclusions.

# 6. Alternative Views

***Predicting measurements is enough, interpretation and decision are unnecessary.*** A plausible view is that models should focus on predicting measurements, and that interpretation and decision-making are unnecessary extras. Our response is that interpretation and decision are essential for clinical meaning and safety: medicine is not just producing numbers, but determining what they imply in context, resolving cross-lead conflicts, and recognizing when evidence is insufficient. So a system that can handle interpretation and decision-making is what makes waveform modeling clinically meaningful and safe.

***End-to-end prediction is sufficient.*** A credible alternative is that end-to-end prediction is enough: prioritize accuracy, calibration, and robustness under shift, and avoid explicitly staged reasoning that can compound errors. Our response is not to replace end-to-end learning, but to constrain it where clinical stakes demand verifiability. The central risk is undetected wrongness under artifacts, missing leads, and device/site heterogeneity, so high average AUROC is not a safety guarantee. Reasoning is therefore warranted only when it is checkable: decisions must be tied to localized time–lead evidence, supported by replayable measurements, challenged by consistency/counterfactual tests, and paired with explicit abstention when evidence is insufficient.

***Data and deployment, not reasoning, are the bottleneck.***
Some researchers argue that progress is constrained by practicalities rather than new framings: better datasets matter more than reasoning. In waveforms, failures are dominated by acquisition noise, device/protocol heterogeneity; without realistic deployment tests, reasoning claims are hard to verify. Our response is that our position is necessary because turning these practical needs into implementable requirements demands a system that couples *waveform semantics* with *language intelligence*: System 1 aligns physiological waveform–language, and System 2 supports agentic Plan–Act–Verify reasoning for complex waveform reasoning.

## 7. Conclusion

We frame physiological waveform reasoning as verifiable, episode-level inference that links raw biosignals to clinical decisions beyond end-to-end prediction. Accordingly, we propose a dual-process blueprint: System 1 provides a stable waveform–language alignment interface that exposes queryable signal evidence, while System 2 performs agentic Plan–Act–Verify reasoning, acquiring missing information and applying deterministic consistency checks under uncertainty. To make progress comparable and deployable, we advocate evaluation through a Verification Ledger that prioritizes traceability and robustness beyond accuracy.

## Acknowledgement

This research was partially supported by the U.S. National Science Foundation under Award Numbers 2442172, 2312502, 2319449, 2211557, 2303037, 2312501, 2425919, 2413417, 2531008, and 2106859, and by the U.S. National Institutes of Health under Award Numbers K25DK135913, RF1NS139325, R01DK143456, U18DP006922, OT2OD038003, R01HL175135, U54OD036472, U54HG012517, and U24DK097771. This research was also partially supported by internal funds and GPU servers provided by the Computer Science Department of Emory University, the SRC JUMP 2.0 Center, Amazon Research Awards, Snapchat Gifts, Optum AI, NEC, the Easton Center, GPU resources from the iTiger GPU cluster (Sharif et al., 2025), and the U.S. National Science Foundation ACCESS program through allocation CIS260707.

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

# A. Common Physiological Waveforms

This appendix briefly recaps representative physiological waveform modalities referenced in the main text.

**Electrocardiogram (ECG).** Records cardiac electrical depolarization and repolarization. It is a quasi-periodic signal (typically 0.05–50Hz) where specific morphological components (P, QRS, T waves) map directly to conduction pathway integrity and myocardial state. Clinical datasets typically consist of 10-second, 12-lead recordings with structured diagnostic annotations (Wagner et al., 2020), often linked to auxiliary clinical data like comorbidities (Johnson et al., 2023).

**Photoplethysmogram (PPG).** Measures peripheral blood volume changes via optical absorption. It encodes hemodynamic parameters including heart rate and vascular tone (typically 0.5–2Hz) and is coupled to the ECG via Pulse Transit Time. PPG quality is highly sensitive to sensor placement, skin tone, and motion artifacts (Bent et al., 2020; Fine et al., 2021), often requiring task-specific window lengths (e.g., shorter for HR, longer for respiratory modulation) (Allen, 2007).

**Electroencephalogram (EEG).** Captures synaptic potentials aggregated across the scalp. It is characterized by spectral dominance across distinct bands—delta ($\sim$0.5–4Hz) to gamma ($>$30Hz)—and transient graphoelements linked to cognitive or pathological brain states. Signal characteristics are heavily influenced by the electrode montage (e.g., 10-20 system) and neural dynamics (Craik et al., 2019).

**Electromyogram (EMG).** Measures muscle motor unit activation via electrical potentials. It reflects neuromuscular recruitment through burst patterns and high-frequency spectral signatures (20–500Hz) (De Luca, 1997). Unlike ECG, EMG lacks a universally standardized montage, often adapting sensor placement to specific muscle groups following guidelines like SENIAM (Hermens et al., 2000).

**Phonocardiogram (PCG).** Records mechanical heart sounds (acoustic vibrations). It validates the mechanical response to electrical excitation, emphasizing valve closure timing (S1/S2) and turbulent flow (murmurs).

**Other physiological waveforms.** Many additional physiological signals appear in modern monitoring, including arterial blood pressure (ABP), respiratory waveforms such as airflow or impedance-based respiration, capnography for end-tidal $CO_2$, electrodermal activity (EDA), and peripheral temperature.

# B. Call to Action for Verifiable Physiological Waveform Reasoning

This appendix translates the paper's position into concrete, stakeholder-specific steps. The goal is to move the community from end-to-end prediction toward verifiable, replayable, budget-aware waveform reasoning, where every decision is supported by localized evidence and reproducible checks.

## B.1. Who should do what

**Benchmark and dataset builders.** (i) Publish benchmarks that mirror deployment: artifacts, missing channels, device/site heterogeneity, and label uncertainty. (ii) Provide an official evaluation harness that consumes a minimal Verification Ledger; phase it in via (a) bonus scoring for ledger outputs, then (b) main-leaderboard requirement. At minimum, the ledger should record: claim type, time/lead span, tool-derived measurements (with units), checker/tool version, and provenance. (iii) Release perturbation generators with fixed seeds and documented ranges, covering noise, baseline wander, motion artifacts, resampling, channel drop, plus at least one deployment-realistic failure mode. (iv) Define tiered audit budgets and protocols that cap inspection time, tool calls, and optional human review, enabling realistic and comparable evaluation.

**Model builders.** (i) Expose structured evidence outputs as first-class predictions, including signal quality, localized events with time and channel indices, and unit-consistent measurements. (ii) Implement a Plan–Act–Verify loop that can request missing evidence, invoke deterministic tools, perform consistency checks, and abstain or escalate when evidence is insufficient. (iii) Report verifiability metrics together with task performance, including traceability to evidence, replay success, robustness under perturbations, and calibrated abstention under budgets.

**Tooling community.** (i) Maintain versioned, deterministic tool suites for quality assessment, delineation, interval measurement, morphology descriptors, and artifact detection. (ii) Enforce provenance logging as part of the tool interface, including tool identity, version hash, parameters, input window, channels, and failure status. (iii) Provide conformance tests and reference outputs so that ledger checks are replayable across platforms and environments.

**Clinical and deployment partners.** (i) Define operational policies for abstention and escalation, specifying what constitutes

sufficient evidence for each workflow setting. (ii) Specify audit budgets and oversight requirements, including what evidence must be shown for acceptance and what triggers human review. (iii) Provide structured feedback traces that identify which evidence was missing, unreliable, or misleading, enabling targeted improvement of tools and models.

### B.2. Benchmark roadmap as a community plan

This roadmap is a staged plan that coordinates multiple stakeholders. Dataset builders provide the benchmark and perturbations, tool developers provide replayable checks, model builders produce ledger-backed decisions.

**Phase 1: Ledger-first leaderboards, months 0 to 6.** (i) Benchmark builders add a ledger requirement to existing tasks and publish a validator that checks schema compliance and replayability. (ii) Tooling community releases a reference tool suite and conformance tests so that reported measurements can be reproduced. (iii) Model builders submit systems that produce structured verification objects, explicit checks, and abstention decisions under stated budgets.

**Phase 2: Artifact and heterogeneity challenge sets, months 6 to 18.** (i) Benchmark builders release controlled perturbation suites and device or site shift splits with documented protocols. (ii) Community evaluates robustness together with sensitivity, focusing on unsafe high-confidence errors and whether abstention reduces such failures under budgets.

**Phase 3: End-to-end workflow episodes, months 18 and beyond.** (i) Benchmark builders publish multi-step episodes that mirror real workflows, including quality gating, evidence extraction, measurement, decision-making, and escalation. (ii) Clinical partners define workflow-specific evidence requirements and escalation policies so success reflects operational safety constraints. (iii) Model builders integrate System 1 evidence extraction with System 2 Plan–Act–Verify, demonstrating replayable reasoning under realistic audit budgets.

## C. Verification Ledger and Verification Objects: Schema and Case Studies

This appendix specifies a minimal, auditable *Verification Ledger* and concrete *verification objects*. The ledger is the evaluation interface: an episode-level conclusion is credited only if it can be re-derived from (i) localized verification objects and (ii) replayable procedures recorded in the ledger.

| **Minimal Verification Ledger fields** | |
|---|---|
| **Inputs** | Record identifier; record hash; sampling rate; lead set; acquisition metadata; task query. |
| **Pointer** | Canonical scope pointer for all evidence: record_id; signal_space raw or preproc; leads; sample_start/sample_end; fs; preproc_id; view_id. |
| **Preprocessing** | preproc_id; ordered ops such as filtering, resampling, normalization; parameters; determinism settings; output hash. |
| **Views** | view_id; scope pointer lead subset and time window; preproc_id; render parameters. |
| **Verification Objects** | Signal-quality summaries; localized events; unit-consistent measurements with units and scope pointer; conflicts with scope and resolution pointers. |
| **Tool Run** | tool name; semantic version; code hash; runtime or container hash; parameters; seed; input pointers; status; failure mode; output hash. |
| **Deterministic Checks** | name; input object ids or pointers; rule; threshold; tolerance; result; tool-run provenance reference. |
| **Decision** | Final label or text; abstain flag; escalate flag; supporting object/check pointers; unresolved conflicts; constraints_checked. |
| **Budget** | Tier id; caps max views, max tool calls, max wall-clock, optional human review; realized usage views, tool calls, wall-clock, audit steps. |

### Verification object and check templates

| | |
|---|---|
| **Pointer** | `{id, record_id, signal_space, leads, s_start, s_end, fs, preproc_id, view_id}` |
| **Preproc** | `{id, input_pointer, ops, params, determinism, output_hash}` |
| **View** | `{id, scope_pointer, preproc_id, render_params}` |
| **Signal Quality Summary** | `{id, metric, value, unit, scope_pointer, threshold, flag, method_ref, provenance_ref}` |
| **Localized Event** | `{id, type, scope_pointer, confidence, attributes, notes, method_ref, provenance_ref}` |
| **Unit Consistent Measurement** | `{id, name, value, unit, scope_pointer, method_ref, uncertainty, tolerance, provenance_ref}` |
| **Tool Run (Provenance)** | `{id, tool, version, code_hash, runtime_env_hash, params, seed, input_pointers, status, failure_mode, output_hash}` |
| **Conflict** | `{id, scope_pointer, sources object_ids, description, severity, resolution_status, resolution_pointers}` |
| **Deterministic Check** | `{id, name, input_object_ids, rule, threshold, tolerance, result, provenance_ref}` |
| **Decision** | `{id, label, text, abstain, escalate, supports, constraints_checked, unresolved_conflicts}` |

### Audit rules: What makes an episode verifiable

| | |
|---|---|
| **Traceability** | Every major claim and the final decision reference explicit pointers to verification objects and deterministic checks. Free-form rationales without pointers are not creditable. |
| **Replayability** | Any quantitative statement is reproducible by re-running the logged tool run using the recorded code hash, runtime hash, parameters, and seed under the recorded scope pointer, matching within the stated tolerance. |
| **Unit and Scope Discipline** | Measurements carry units and an explicit scope pointer lead and time in sample indices plus fs and method_ref. Ambiguity in unit, scope, or method breaks replayability. |
| **Determinism Discipline** | All tools that can affect values must log code hash, runtime or container hash, and seed; checks must log tolerance. Missing determinism metadata downgrades the episode to non-verifiable. |
| **Quality Gating** | If a quality summary fails for a required scope pointer, the episode must query an alternative scope, abstain or escalate, or log a deterministic check that justifies validity under the limitation. |
| **Conflict Handling** | If conflicts are detected across tools, views, or scopes, the ledger logs scope, sources, and resolution pointers. Unresolved high-severity conflicts cannot support a definitive decision. |
| **Budget Compliance** | Credited decisions must satisfy the declared tier caps for views, tool calls, and wall-clock; budget overruns are logged and may be disqualified per protocol. |

## C.1. Case Study

This case study illustrates uncertainty-aware safety in a verifiable episode. The system first records objective signal-quality evidence (e.g., lead dropout and low SNR) as verification objects tied to the evaluated time/lead scope. It then runs two independent rhythm analyses whose outputs disagree at high severity (AF vs. sinus), and logs this disagreement as an unresolved conflict object. Two deterministic checks—(i) a quality-gating rule requiring required leads and SNR above a threshold, and (ii) a conflict policy that defers when high-severity disagreements remain unresolved—are applied to these objects. Because the quality gate fails and the conflict policy triggers deferral, the episode returns an explicit abstain+escalate decision ("unknown rhythm"), with a support list that points back to the quality metrics, the conflict, and the specific checks that forced deferral. This makes the safety behavior auditable: a reviewer can replay the checks and verify that abstention follows directly from recorded evidence under the stated policy and budget.

## Case Study: Low-quality recording triggers abstention

| | |
|---|---|
| **Inputs** | Record **rec_789** sampled at **500 Hz**. Leads: I, II, III, aVR, aVL, aVF, V1–V6. Task: determine rhythm and triage. |
| **View** | Single view over the full **10-second** segment (all leads), using the default preprocessing. |
| **Tools executed** | A signal-quality module, two independent rhythm analysis tools (A and B), and a ledger checker. |
| **Quality evidence** | **Lead I dropout detected** $\Rightarrow$ quality failure.
Estimated **SNR = 6 dB** (required $\geq$ **10 dB**) $\Rightarrow$ low-quality flag. |
| **Rhythm hypotheses** | Tool A indicates **atrial fibrillation (AF)**; Tool B indicates **sinus rhythm**.
These outputs form a **high-severity unresolved disagreement**. |
| **Deterministic checks** | **Quality gating**: required leads present and SNR $\geq$ 10 dB $\Rightarrow$ **FAIL**.
**Conflict policy**: unresolved high-severity disagreement $\Rightarrow$ **DEFER**. |
| **Decision** | Return **unknown rhythm**; **abstain** from definitive rhythm labeling and **escalate** for further review.
Supports: lead dropout, low SNR, AF vs sinus disagreement, quality-gating failure, conflict-policy deferral. |
| **Budget** | Tier: standard. Limits: **1 view**, **4 tool calls**, **2 s** wall-clock.
Usage: 1 view, 4 tool calls, 0.9 s wall-clock. |

*Note:* Detailed object identifiers, hashes, and runtime metadata follow Appendix C and are omitted here for readability.

