# OpenReview forum: "Position: Beyond Prediction: Toward Verifiable Physiological Waveform Reasoning with Foundation Models and Agentic LLMs"
_ICML.cc/2026/Position_Paper_Track — ICML 2026 Position Paper Track regular_

### Official Review · Reviewer_1CR1 · 2026-03-12

**Significance:** 2
**Argument Clarity:** 3
**Rating:** 4
**Confidence:** 2

**Questions:**

Regarding Empirical Feasibility and Robustness Verification:
Could the authors provide a minimal empirical Proof-of-Concept (PoC) or a focused case study using a subset of a public dataset (e.g., PTB-XL)?
Specifically, it would be highly convincing to see how the system behaves under intentional signal corruption (such as lead dropout or severe baseline wander).

**Alternative Views Section:**

Yes

**Compliance With Llm Reviewing Policy A Conservative:**

Affirmed.

**Discussion Potential:**

2

**Paper Summary:**

This position paper advocates for a paradigm shift in clinical physiological waveform modeling from black-box "end-to-end prediction" to "verifiable physiological waveform reasoning" to address the critical lack of traceability and safety in current models.
To achieve this, the authors propose a cognitive-science-inspired dual-process architecture:
System 1 handles multimodal waveform-language alignment, while System 2 utilizes an agentic LLM framework (comprising a Curator, Analyst, and Critic) to execute a multi-turn "Plan-Act-Verify" loop via external deterministic tools.
Furthermore, the paper introduces a "Verification Ledger" evaluation mechanism, urging the community to move beyond scalar accuracy metrics toward traceable evidence and uncertainty-aware safe abstention in high-stakes clinical settings.

**Position:**

Yes

**Position In Title:**

Yes

**Related Work:**

2

**Strengths And Weaknesses:**

Strengths:

The paper presents an ambitious conceptual framework for physiological signal modeling. By advocating a shift from “end-to-end black-box prediction” to verifiable waveform reasoning, the authors highlight an important challenge in clinical AI—namely, the need for traceability and safety.

The proposed Verification Ledger evaluation framework is particularly interesting and potentially actionable. By requiring models to provide localized and traceable evidence and enabling safe abstention, the framework addresses an important trust barrier that currently limits the adoption of medical foundation models in clinical environments.

Weaknesses:

1. Lack of Empirical Proof-of-Concept (PoC)

While the proposed dual-system architecture is conceptually well motivated, the manuscript currently does not include even a minimal empirical proof-of-concept (PoC) demonstration.

In particular, System 2 relies on a multi-turn Plan–Act–Verify reasoning loop involving repeated LLM invocations and external tool calls. Such a pipeline may introduce additional computational latency and engineering complexity. Without empirical evaluation, it remains unclear whether the proposed trade-off—between increased verifiability and computational overhead—would be practically feasible in real-world clinical environments, especially in time-sensitive scenarios such as ICU monitoring.

A small-scale prototype experiment or simulation study could significantly strengthen the manuscript by illustrating the practical feasibility of the framework.

2. Domain-specific evaluation and citation issues.

As a reviewer without deep clinical domain expertise, I find it difficult to evaluate how the proposed physiological waveform reasoning system differs from, or improves upon, generic AI-generated solution strategies that could be produced for this type of problem. Additionally, there appears to be a citation issue: Orphanidou et al. (2024a) and Orphanidou et al. (2024b) appear to refer to the same paper.

Overall, the paper presents an interesting conceptual direction for improving the safety and interpretability of clinical AI systems. However, incorporating at least a minimal empirical validation would significantly strengthen the impact and credibility of the proposed framework.

**Support:**

2

---

> ### Author Rebuttal · Authors · 2026-03-30
>
> We sincerely thank you for your thoughtful and constructive feedback. We greatly appreciate your positive assessment of our paper, as well as your helpful suggestions for strengthening it. We have carefully considered your comments and addressed each concern individually in our point-by-point responses.
>
> > **W1: While the proposed dual-system architecture is conceptually well motivated, the manuscript currently does not include even a minimal empirical proof-of-concept (PoC) demonstration.**
>
> > **Q1: Regarding Empirical Feasibility and Robustness Verification: Could the authors provide a minimal empirical Proof-of-Concept (PoC) or a focused case study using a subset of a public dataset (e.g., PTB-XL)? Specifically, it would be highly convincing to see how the system behaves under intentional signal corruption (such as lead dropout or severe baseline wander).**
>
> Thank you for this important suggestion. We agree that a minimal empirical proof-of-concept can make the practical feasibility clearer. We implemented a small PoC using 10 ECG recordings from MIMIC-IV-ECG, including 5 atrial fibrillation cases and 5 sinus rhythm cases. For each record, we constructed three reasoning episodes: clean, hard lead dropout, and severe baseline wander. We compared two settings: (1) a single-shot baseline that only analyzes the initial fixed view (Lead II), and (2) a minimal Plan–Act–Verify prototype. If the initial evidence is insufficient, the prototype is allowed exactly additional deterministic verification actions, such as switching to a backup lead or re-running a measurement pass, after which it must either issue a decision or abstain. This PoC is intentionally simplified and is not meant to represent the full proposed agentic stack. Instead, it isolates the core closed-loop verification idea using lightweight deterministic signal-processing tools, including signal-quality checking, R-peak detection, and RR-irregularity measurement.
>
> In this minimal study, the single-shot baseline achieved 20/30 accuracy overall, whereas the Plan–Act–Verify prototype achieved 27/30. Under clean episodes, both methods performed strongly. Under hard lead dropout, the fixed initial view often became unusable for the single-shot baseline, whereas the prototype was able to recover several cases by invoking one additional verification step on an alternative view. Under severe baseline wander, the baseline also degraded, while the prototype again improved robustness by performing an extra deterministic verification pass when the initial evidence was unreliable. The additional cost remained small: the baseline required 0.107 s per episode on average, whereas the Plan–Act–Verify prototype required 0.189 s, with 1.33 average tool calls per episode.
>
> This study provides a meaningful feasibility signal for the central claim: closed-loop verification can be practically instantiated, can improve robustness under representative nuisance corruptions, and can do so under a limited verification budget.
>
>
> > **W2: As a reviewer without deep clinical domain expertise, I find it difficult to evaluate how the proposed physiological waveform reasoning system differs from, or improves upon, generic AI-generated solution strategies that could be produced for this type of problem.**
>
> Thank you for this thoughtful comment. We agree that, especially for readers without deep clinical expertise, it is important to clarify how our proposal differs from a generic AI-generated problem-solving pipeline. Our intended distinction is that the proposed framework is waveform-specific at multiple levels, rather than simply applying a general LLM/agent recipe to a biomedical task.
> 1. System 1 is designed for physiological waveform-language alignment, with four routes that preserve waveform-grounded information rather than relying only on generic text or multimodal prompting.
> 2. System 2 is centered on waveform-grounded verification, including targeted evidence acquisition, deterministic physiological measurement, consistency checking, and abstention under uncertainty, rather than generic tool use.
> 3. The reasoning objectives are waveform-specific, spanning evidence, interpretation, and decision, and include tasks such as signal quality assessment, event segmentation, physiological measurement, morphology semantics, cross-lead concordance, and guideline-constrained inference.
> 4. The evaluation is also domain-specific: we propose a Verification Ledger and episode-level criteria such as traceability, replayability, robustness, and uncertainty-aware safety, rather than final-answer accuracy alone.
>
>
> > **W3: Additionally, there appears to be a citation issue: Orphanidou et al. (2024a) and Orphanidou et al. (2024b) appear to refer to the same paper.**
>
> Thank you for pointing this out. This is a citation labeling typo in the current draft. We will correct the duplicated reference in the revised version.

---

> > ### Author Rebuttal · Reviewer_1CR1 · 2026-04-02
> >
> > Thank you for your response. As I do not have sufficient familiarity with the background of this work, I am not in a position to provide a fully fair assessment. Please refer to the other reviewers’ comments for a more informed and expert evaluation.

---

### Official Review · Reviewer_TdDb · 2026-03-12

**Significance:** 4
**Argument Clarity:** 4
**Rating:** 6
**Confidence:** 3

**Questions:**

My main question to the authors is: can you elucidate more the verification setup you intend for these systems? What challenges do you foresee in the development of them, and what technologies (if any) would need to be developed in order to aid in the robust verification of these physiological reasoning systems?

**Alternative Views Section:**

Yes

**Compliance With Llm Reviewing Policy A Conservative:**

Affirmed.

**Discussion Potential:**

4

**Final Justification:**

I remain convinced of my original assessment.

**Paper Summary:**

The authors argue for physiological monitoring AI systems to explicitly incorporate reasoning for the purposes of explainability and external verification. They define a roadmap towards such systems by defining how the foundation models (system 1) should be leveraged alongside reasoning systems (system 2) to accomplish various clinically-relevant endpoints in physiological monitoring. The authors claim that by employing evidence-based reasoning, these systems can be used with higher confidence in critical application. They then outline 4 high level challenges to building such systems which must be addressed, largely related to domain-specific data challenges. They proceed to argue to system 1 alignment between natural language and physiological alignment, before training a system 2 “plan-act-verify” agentic reasoning loop on top of this foundation to produce verifiable reasoning chains. The paper closes with a set of verification schemes for such a system, and 3 contrasting views to their position.

**Position:**

Yes

**Position In Title:**

No

**Related Work:**

4

**Strengths And Weaknesses:**

In my estimation (aside from the title needing rephrasing) this is a very strong submission to the position paper track. It lays out a clear roadmap to a modern, agentic AI system applied to a meaningful domain where implementation risks are high and the potential utility to society is large. The individual elements of the paper are well referenced and ambitious - to build such a system properly is a substantial scientific challenge that will require the concentrated effort of many researchers in collaboration, and as such, the position paper is a meaningful contribution to the field to synchronize the debate around a concrete framework. In particular, although the paper focuses on a domain that would be considered niche by many ML researchers, the challenges referenced and addressed here are present in many AI for Clinical Support domains, and raises a reasonable strategy that should be considered in all of these domains. In light of this, part of me wishes the paper was written for a broader audience, but I think the claims are easier to follow in a grounded application and researchers in other fields will easily be able to build the mapping to their own problem. My main critique of the paper revolves around the verification strategy discussed (“code as reasoning”) - it is not clear to me how the authors intend to verify the reasoning of the system, which is a crucial detail for their plan, but perhaps this will be a focus of upcoming research. In any case, in the areas where the researchers are unclear on the exact verification methods to explore, it would be better to make them explicit in the position paper so that researchers can quickly identify it as an area in need of focused attention.

**Support:**

4

---

> ### Author Rebuttal · Authors · 2026-03-30
>
> We sincerely thank you for your thoughtful and constructive feedback. We greatly appreciate your positive assessment of our paper, as well as your helpful suggestions for strengthening it. We have carefully considered your comments and addressed each concern individually in our point-by-point responses.
>
> > **W1: The title needing rephrasing**
>
> Thank you for this suggestion for the title. We will revise it to better reflect both the central thesis—moving beyond prediction-centric pipelines toward verifiable physiological waveform reasoning—and the proposed dual-system perspective. We therefore plan to revise it accordingly: Position: Beyond Prediction: Toward Verifiable Physiological Waveform Reasoning with a Dual-System Framework.
>
> We would also appreciate any further suggestion from you on title phrasing.
>
>
> > **W2: My main critique of the paper revolves around the verification strategy discussed (“code as reasoning”) - it is not clear to me how the authors intend to verify the reasoning of the system, which is a crucial detail for their plan, but perhaps this will be a focus of upcoming research. In any case, in the areas where the researchers are unclear on the exact verification methods to explore, it would be better to make them explicit in the position paper so that researchers can quickly identify it as an area in need of focused attention.**
> > **Q1: Can you elucidate more about the verification setup you intend for these systems?**
>
>
> Thank you for this thoughtful comment. We would like to clarify that “code as reasoning” in our paper refers to using deterministic tools or executable procedures for quantitative waveform claims (e.g., measurements or consistency checks), so that intermediate results are reproducible and auditable.
>
> More broadly, verification in our framework is a closed-loop Plan–Act–Verify process: the Curator acquires missing verification objects, the Analyst performs deterministic measurement or tool-grounded execution, and the Critic checks physiological or guideline consistency, resolves contradictions via backtracking, and triggers abstention or escalation when support is insufficient. The overall intent is not to claim that a single finalized verification algorithm already exists, but to make the verification contract explicit: reasoning should be grounded in auditable verification objects, recorded in a replayable Verification Ledger, and evaluated at the episode level rather than by final-answer accuracy alone.
>
> We will revise the paper to better distinguish the specified framework components from the concrete verification methods that remain an important direction for future work.
>
>
>
> > **Q2: What challenges do you foresee in the development of them, and what technologies (if any) would need to be developed in order to aid in the robust verification of these physiological reasoning systems?**
>
>
> Thank you for this important question. We agree that this point should be made more explicit. In our view, the main challenges are: (1) reliably extracting verification objects under acquisition heterogeneity, signal artifacts, and missing or corrupted channels; (2) overcoming the data-reasoning mismatch, since most current datasets do not provide intermediate evidence, rule traces, or audit-ready supervision; and (3) designing closed-loop reasoning policies that know when to re-measure, backtrack, abstain, or escalate under uncertainty. These are precisely the bottlenecks that motivate our shift from prediction-centric systems to verifiable waveform reasoning.
>
> The technologies we believe are most needed are correspondingly concrete: (1) stronger waveform-language alignment mechanisms that preserve measurable signal evidence; (2) deterministic tool suites for delineation, interval/quality measurement, and robustness checks; (3) physiological/guideline constraint modules for consistency verification; and (4) the replayable Verification Ledger that records evidence objects, tool provenance, and final decisions. In our framework, these components are operationalized through System 1 for waveform-language alignment and System 2 for Plan-Act-Verify reasoning with Curator/Analyst/Critic roles, and are evaluated through traceability, replayability, robustness, and uncertainty-aware abstention rather than final-answer accuracy alone.

---

> > ### Author Rebuttal · Reviewer_TdDb · 2026-04-03
> >
> > Thank you for your detailed reply!

---

### Official Review · Reviewer_DKxA · 2026-03-13

**Significance:** 4
**Argument Clarity:** 3
**Rating:** 5
**Confidence:** 4

**Questions:**

NA

**Alternative Views Section:**

Yes

**Compliance With Llm Reviewing Policy A Conservative:**

Affirmed.

**Discussion Potential:**

3

**Paper Summary:**

This paper argues that for physiological signals (e.g., ECG, PPG, EEG) should not be performing end-to-end prediction, but should be used for reasoning system to drive faithful clinical decisions. Authors first performed limitation analysis on why current foundation models are insufficient. Authors later proposed a dual-process framework, which first aligns physiological signals with natural languages, and later using AI agents to perform verifiable physiological signal reasoning. The paper finally proposed evaluations methods beyond end-to-end prediction, providing verifiable and episode-level decision making that is more friendly in clinical settings.

**Position:**

Yes

**Position In Title:**

Yes

**Related Work:**

3

**Strengths And Weaknesses:**

**Strengths**
- Complete and thorough literature review that shows the current limitations of existing foundation models and systems, which provided sufficient amount of reasoning for the paper's position. The paper also clearly listed possible reasoning structures that are currently common in LLM systems that can be applied on physiological systems.
- Each argument made in this paper is well supported by examples and reasoning.
1. The paper clearly stated the limitations of current models, by both stating the limitation from the signal perspective and also the model perspective.
2. The proposed framework has clear design strategies with various design choices. The authors proposed four alignment routes that links physiological systems with natural languages, that suits different types of model architectures. The Plan–Act–Verify pipeline is also clearly stated and motivated.

**Weaknesses**
- The authors can potentially state the potential drawback and limitations of the proposed approach, such as physiological semantic/information fidelity, risks on involving additional human/expert labors, also potentially increased training costs.

**Support:**

3

---

> ### Author Rebuttal · Authors · 2026-03-30
>
> We sincerely thank you for your thoughtful and constructive feedback. We greatly appreciate your positive assessment of our paper, as well as your helpful suggestions for strengthening it. We have carefully considered your comments and addressed each concern individually in our point-by-point responses.
>
> > **W1: The authors can potentially state the potential drawback and limitations of the proposed approach, such as physiological semantic/information fidelity, risks on involving additional human/expert labors, also potentially increased training costs.**
>
>
> Thank you for this important suggestion. We agree that the current manuscript would benefit from stating the practical limitations of the proposed framework.
>
> Regarding physiological semantics and information fidelity: these issues are central motivations of our framework. In the paper, System 1 is designed to align physiological waveforms with language so that reasoning remains grounded in waveform semantics, while System 2 performs Plan-Act-Verify reasoning with deterministic measurements, consistency checks, and abstention under uncertainty.
>
> We agree that human/expert oversight and additional system costs are real trade-offs. Our intent is not to require expert review for every case, but to enable targeted audit and escalation in safety-critical or evidence-conflicting scenarios. Likewise, compared with single-shot prediction, closed-loop reasoning may incur extra tool calls, longer trajectories, and higher engineering/training/deployment cost.
>
> To make these trade-offs explicit, we will add the following brief limitation discussion in the revised version:
>
> As a limitation, the proposed framework may still depend on targeted human or expert oversight in difficult cases, especially when signal quality is poor, evidence is conflicting, or the system abstains. This improves safety and auditability, but also introduces additional labor and workflow burden. Moreover, compared with single-shot end-to-end prediction, the closed-loop Plan-Act-Verify design may incur higher system cost, including extra tool usage, longer reasoning paths, greater engineering complexity, and potentially increased training and inference overhead. These trade-offs are important to acknowledge when considering practical deployment.

---

> > ### Author Rebuttal · Reviewer_DKxA · 2026-04-03
> >
> > I am curious if the authors have considered the data aspect of the proposed position. Which existing datasets can support the proposed framework, what type of datasets are required to support the proposed framework (both modality-wise and quantity/quality-wise)?

---

> > > ### Author Response · Authors · 2026-04-05
> > >
> > > Thank you for this very helpful follow-up. We do consider the data aspect explicitly in the paper. In **Section 3.1, Challenge 4 (“The Data-Reasoning Mismatch”)**, we explain that most existing public datasets provide coarse labels or reports, but lack intermediate evidence objects, rule traces, tool provenance, and episode-level supervision. We further discuss the dataset and benchmark implications in **Appendix B.1 (“Benchmark and dataset builders”)** and **Appendix B.2 (“Benchmark roadmap as a community plan”)**, where we outline the kinds of datasets and evaluation interfaces needed to support verifiable waveform reasoning.
> > >
> > > Existing datasets only partially support the proposed framework. Public waveform datasets, especially ECG datasets such as MIMIC-IV-ECG, can support parts of the early-stage System 1 alignment, and component-level evaluation. Clinically oriented benchmarks such as ECG-Expert-QA are also useful for evaluating aspects of waveform understanding and reasoning. However, these resources remain insufficient for full verifiable waveform reasoning, because most provide waveform–label pairs, waveform–report pairs, or QA-style supervision, rather than the structured intermediate evidence needed for replayable, ledger-backed reasoning.
> > >
> > > Modality-wise, ECG is currently the most mature setting because it already has relatively large public corpora, and paired reports in some cases. Other modalities such as PPG, EEG, EMG, and PCG can support parts of the framework, but are generally less mature for full reasoning because paired language supervision, standardized evidence targets, and replayable evaluation interfaces remain limited. In terms of data quantity and quality, we believe the key bottleneck is not raw scale alone, but the availability of high-quality evidence-centric data, including localized events, unit-consistent measurements, perturbation protocols, provenance-aware tool outputs, and workflow-style episode annotations. This is why the paper calls for a staged roadmap from ledger-first benchmarks, to artifact and heterogeneity challenge sets, and ultimately to multi-step workflow episodes for end-to-end verifiable reasoning.

---

### Decision · Program_Chairs · 2026-04-30

**Decision:**

Accept (regular)

**Comment:**

This paper advocates shifting from black-box prediction to verifiable physiological waveform reasoning in clinical AI. It proposes a dual-process architecture integrating foundation models with a Plan-Act-Verify agentic loop, evaluated via a novel Verification Ledger.
The submission is well-motivated and ambitious. Reviewer DKxA praised the clear design strategies and thorough literature review. Reviewer TdDb highlighted the work as a clear roadmap for agentic AI applied to a high-risk domain. Reviewer 1CR1 found the Verification Ledger to be an actionable framework for improving traceability.
Initial weaknesses included the lack of an empirical proof-of-concept (1CR1) and a need for deeper discussions on data requirements and verification limits (DKxA, TdDb). The authors' rebuttal successfully provided a minimal proof-of-concept using MIMIC-IV-ECG and clarified systemic constraints.